# Correlation of Blood Hemoglobin Values with Non-Invasive Co-Oximetry Measurement of SpHb in Dogs Undergoing Elective Ovariohysterectomy

**DOI:** 10.3390/ani14060822

**Published:** 2024-03-07

**Authors:** María Fernanda Espinosa-Morales, Agatha Elisa Miranda-Cortés, Daniel Mota-Rojas, Alejandro Casas-Alvarado, Alejandro Jiménez-Yedra, Alicia Pamela Pérez-Sánchez, Ismael Hernández-Ávalos

**Affiliations:** 1Clinical Pharmacology and Veterinary Anesthesia, Biological Sciences Department, Facultad de Estudios Superiores Cuautitlán, Universidad Nacional Autónoma de Mexico (UNAM), Cuautitlán 54714, Mexico; 2Neurophysiology of Pain, Behavior and Animal Welfare Assessment, Department of Agricultural and Animal Production (DAAP), Universidad Autónoma Metropolitana (UAM), Xochimilco Campus, Mexico City 04960, Mexico; 3Department of Anesthesiology in Dogs and Cats, Specialty Veterinary Hospital, Facultad de Medicina Veterinaria y Zootecnia (FMVZ), UNAM, Mexico City 04510, Mexico; 4Veterinary Hospital for Dogs and Cats, Specialty in Medicine and Surgery in Dogs and Cats, Facultad de Medicina Veterinaria y Zootecnia (FMVZ), Universidad Popular Autónoma del Estado de Puebla (UPAEP), Puebla 72410, Mexico

**Keywords:** anesthetic monitoring, hemoglobin, SpHb, dogs

## Abstract

**Simple Summary:**

The addition of pulse co-oximetry in anesthesia monitoring to continuously determine blood hemoglobin concentration (SpHb) represents a biomarker correlating with the oxygen saturation of patients undergoing surgery. This study aimed to correlate and determine the measurement bias between blood Hb levels with SpHb and arterial oxygen content (SpOC), both derived from pulse co-oximetry in dogs undergoing elective ovariohysterectomy (OVH). Eighty-five bitches were admitted to elective surgery where SpHb was measured during surgery. Five minutes before the end of the procedure, blood sampling was performed to determine the blood Hb concentration (Hb_LAB_). The Bland–Altman analysis showed 95% limits of agreement from −4.22 to 4.99 g/dL, and a positive correlation was found between SpHb and Hb_LAB_ (r = 0.401). In conclusion, SpHb presents a moderate positive correlation with direct Hb blood concentration. This possibly shows that continuous measurement of SpHb by noninvasive co-oximetry is a reliable and advanced alternative for monitoring Hb concentration in dogs under anesthesia.

**Abstract:**

Cardiovascular function monitoring has been suggested as a key parameter to determine patient stability during the anesthetic process. However, the use of pulse co-oximetry has been suggested as a technology to complement the monitoring of this system as a direct way to assess hemoglobin (Hb) blood concentration. Therefore, this study aimed to correlate and determine the measurement bias between Hb blood levels with continuously determined blood hemoglobin concentration (SpHb) and arterial oxygen content values (SpOC), both obtained by noninvasive co-oximetry in dogs undergoing elective ovariohysterectomy (OVH). A total of 85 clinically healthy bitches of different breeds that were admitted for elective OVH surgery were evaluated. These animals underwent SpHb and SpOC capture after the in vivo setting for the duration of the surgical procedure. Likewise, five minutes before the end of the surgical procedure, a blood sample was obtained directly from the jugular vein to determine the blood concentration of Hb (Hb_LAB)_. The Bland–Altman analysis showed 95% limits of agreement from −4.22 to 4.99 g/dL with a BIAS (mean difference) of 0.384 ± 2.35 g/dL (r = 0.401). SpHb recordings were correlated with oxygen saturation (SpO_2_) (r = 0.995), SpOC (r = 0.992) and with perfusion index (PI) (r = 0.418). Therefore, SpHb presents a moderate positive correlation with direct blood concentration of Hb. This possibly shows that continuous measurement of SpHb by noninvasive co-oximetry is a reliable and advanced alternative for monitoring Hb concentration in dogs under anesthesia.

## 1. Introduction

General anesthesia is often associated with cardiorespiratory risks due to the inherent effects of the anesthetic agents or to individual idiosyncrasy. Although the risk can be reduced with a pre-anesthetic clinical assessment, veterinary patients are not exempt from presenting cardiorespiratory alterations. Perianesthetic monitoring of the cardiovascular function is essential to promptly recognize variations in vital functions that may compromise homeostasis. It can also help to install corrective therapies, which increase anesthetic safety, reduce morbidity/mortality, and prevent complications during the perioperative period [1,2].

Pulse oximetry is commonly evaluated during the perioperative period. Pulse oximetry is a non-invasive technique that determines in real-time the oxygen saturation associated with hemoglobin (Hb) in arterial blood [3]. This is performed through the assessment of the absorbance level of red light where deoxyhemoglobin absorbs more red light than oxyhemoglobin, while infrared light estimates the saturation of O_2_ detected in the pulse (SpO_2_) by determining the absorbance difference [4,5]. Due to the importance of Hb in gas exchange, the evaluation of Hb has been adopted as a way of estimating the stability level during the transoperative period in patients under anesthesia [6]. In relation to this, O2 is transported dissolved in solution in <2% and combined with Hb in >98%, so in the anesthetic–surgical patient or those with critical illnesses, the evaluation and determination of blood Hb is very important to prevent or control states of anemia and hypoxia, especially in those individuals where the risk of hemorrhage may be high [7].

Blood Hb is a parameter that can also help estimate the correct ventilation process when it is correlated with other cardiorespiratory parameters such as the inspired fraction of oxygen (FiO_2_) and the end-tidal CO_2_ (ETCO_2_), since this protein is the main component with which oxygen binds [8,9]. This indicator can also provide precise information on the level of lost blood components during the surgical period [8]. In a clinical setting, changes in Hb concentration can also be an indicator of alterations in the oxygen supply in peripheral tissues. Moreover, during surgical procedures or in intensive care units, Hb concentrations can serve as a guide for deciding on when to perform blood transfusions, showing the importance of Hb monitoring during the anesthetic–surgical period [10,11,12].

Currently, the standard method for determining Hb levels is direct measurement of methemoglobin cyanide, which can directly diagnose anemia and is the main determinant of the oxygen transport capacity to tissues [13,14,15]. However, this invasive method requires blood sampling and, regardless of its accuracy, it has limitations such as the cost and need for skilled personnel [13,14,16].

Pulse co-oximeters use transcutaneous technology based on multi-wavelength spectrophotometry for a continuous determination of hemoglobin (SpHb) among other parameters such as oxygen saturation derived from pulse co-oximetry (SpO_2_), arterial oxygen content derived from pulse co-oximetry CaO_2_ (SpOC), methemoglobin (SpMet), carboxyhemoglobin (SpCO), perfusion index (PI), and the plethysmography variability index (PVi) [13,14,17,18].

Devices that continuously and non-invasively measure SpHb help in clinical decision making regarding the needs of each patient [13,14]. Comparative studies evaluating the accuracy of these non-invasive technologies and laboratory methods (e.g., arterial blood gas measurement or point-of-care (POC) blood testing) have been published [19,20]. These studies emphasize the importance and application of this parameter. However, in veterinary medicine, limited information on SpHb assessment through non-invasive co-oximetry in dogs undergoing surgery has been published.

This study aimed to correlate and determine the measurement bias between blood Hb levels and SpHb obtained by noninvasive co-oximetry after in vivo adjustment in dogs undergoing elective ovariohysterectomy (OVH). We hypothesized that the Hb measured via pulse co-oximetry acceptably correlates with the laboratory measurement with the advantage of continuous monitoring.

## 2. Materials and Methods

### 2.1. Animals

Eighty-five clinically healthy dogs admitted for elective ovariohysterectomy were selected for the present study. The sample size was calculated using G*Power 3.1.9.7 (Heinrich-Heine-Universität Düsseldorf, Düsseldorf, Germany) considering a power of 0.9 (1−β error) and an α error of 5%. Written informed consent signed by the owner was obtained before the surgical procedure. All animals underwent a complete blood cell count, serum biochemistry, and urinalysis three days before surgery. On the day of surgery, the animals went through a complete physical examination. Only animals classified as ASAI or ASAII according to the American Society of Anesthesiologists [21] were included in the study. As exclusion criteria, dogs in estrus, animals with anemia, coagulation disorders, kidney or liver disease, or pyometra were not admitted.

### 2.2. Surgical Anesthetic Procedure

All animals were fasted from food and water six hours before anesthetic induction. During the pre-anesthetic preparation, the forelimbs and the surgical area were shaved with a machine and a number 40 blade (Andis AGC2 Ultraedge, Mexico City, Mexico). In the right forelimb, a 22G × 19 mm caliber catheter was placed aseptically in the cephalic vein to administer Hartmann solution at an infusion rate of 5 mL/kg/h (BeneFusion VP1 Vet, Mindray, Hamburg, Germany) throughout the surgical procedure [22]. In the left forelimb, blood pressure was evaluated through non-invasive oscillometry (NIBP) over the radial artery using the most adequate size of blood pressure cuff (Vet25, SunTech Medical, Morrisville, NC, USA). Using this method, systolic blood pressure (SAP) and diastolic blood pressure (DAP) were obtained, expressed in millimeters of mercury (mmHg). Mean arterial pressure (MAP) was obtained by the following correction formula: MAP = [(SAP − DAP) ÷ 3) + DAP] [23]. Subsequently, dexmedetomidine hydrochloride was administered at a sedative dose of 2 µg/kg intravenously (IV) (Dexdomitor, Zoetis, Mexico City, Mexico); after 10 min, meloxicam at 0.2 mg/kg IV (Metacam, Boehringer Ingelheim, Mexico City, Mexico) and tramadol hydrochloride at 4 mg/kg IV (Pisadol, PiSA, Mexico City, Mexico) were administered.

After 20 min, anesthetic induction was performed with propofol (Recofol, Pisa, Mexico City, Mexico) at 2–3 mg/kg. Once an adequate state of unconsciousness was reached, the tracheal tube was connected to an anesthetic rebreathing circuit with an oxygen flow of 45 mL/kg/min. Anesthesia was maintained with isoflurane (Sofloran, PiSA, Mexico City, Mexico) vaporized in 100% oxygen while adjusting the vaporizer dial initially to 1.7% and then modifying the concentration to maintain adequate surgical anesthetic depth [24]. The depth of surgical anesthesia was maintained with an End Tidal of isoflurane (ET_ISO_) of 1.57 ± 0.37% (ePM12VETc/AA, Mindray, Hamburg, Germany) monitoring the relaxation of the mandibular tone, ventromedial deviation of the eyeball and absence of palpebral reflex. During the anesthetic–surgical procedure, the dogs were mechanically ventilated with the ventilator integrated into the anesthesia station (Wato-EX20 vet, Mindray, Hamburg, Germany), using a pressurometric controlled ventilation (PCV) method at a mean pressure of the airway (Paw) of 10–15 cmH_2_O and a respiratory rate of 15–20 breaths per minute during surgery to maintain an End Tidal CO_2_ (ETCO_2_) of 35–45 mmHg.

Thermal support was provided to the patients for trying to maintain normothermia during the surgery, maintaining the temperature between 36–38 °C (Equator^®^, EQ-5000, Smiths Medical ASD Inc., Minneapolis, MN, USA). Intraoperative body temperature was evaluated using an esophageal thermometer (ePM12VETc/AA, Mindray, Hamburg, Germany). Before anesthesia, rectal temperature was obtained using a digital thermometer (neutek, MT 201, Ningbo, China). All surgeries were performed by the same surgeon using a midline approach and a triple hemostatic surgical technique. The anesthetic procedures were performed by the same anesthesiologist.

### 2.3. Blood Hemoglobin (Hb_LAB_) and Continuous Determination of Total Hemoglobin Derived from Pulse Co-Oximetry (SpHb)

The present study was a prospective clinical trial. Once the animals were in the anesthetic–surgical plane, the adhesive sensor (R125L, SpHb neonatal, Masimo, Irvine, CA, USA) of the Radical 7 pulse co-oximeter (Rainbow SET, Masimo, Irvine, CA, USA) was placed on the ventrolateral portion of the tongue (Figure 1) to measure total hemoglobin (SpHb) and arterial oxygen content (SpOC). The evaluation of these parameters began once the in vivo adjustment was performed, 15 min after the first reading [13,14]. The in vivo calibration of the continuous and non-invasive SpHb measurement device was carried out considering an automatic calibration factor obtained from the subtraction of the value of the same SpHb from the value of total blood hemoglobin measured at the same time (15 min after the first reading was recorded).

Both SpHb and SpOC were evaluated every five minutes after the in vivo adjustment during the intraoperative period. To determine the correlation of the hemoglobin measurement methods, five minutes before finishing the surgery, a blood sample of 3 mL was taken in a tube with EDTA from the jugular vein. The sample was correctly identified and processed in an automated analyzer for blood biometry (Lifotronic AC6010 Vet, Shenzhen, China) for direct hemoglobin (g/dL) determination (Hb_LAB_).

### 2.4. Monitoring of Cardiorespiratory Parameters during Anesthesia

During the anesthetic–surgical procedure, cardiorespiratory parameters were evaluated at baseline (T_BASAL_), sedation (T_DEXME_), anesthetic induction (T_INDUC_), five minutes after anesthetic induction (T_5_), 10 min after anesthetic induction at the beginning of surgery (T_10_), and every five minutes until the end of the procedure (T_15_, T_20_, T_25_, T_30_, T_35_). Heart rate (HR), respiratory rate (RR), and systolic (SAP), diastolic (DAP), and mean blood pressure (MAP) were evaluated through non-invasive oscillometry (NIBP) on the left forearm above the radial artery using the most adequate size of blood pressure cuff. Temperature (T°) and SpO_2_, were evaluated through a multiparameter monitor (ePM12VETc/AA, Mindray, Hamburg, Germany).

The PI and PVi indices were evaluated by infrared reading (Radical 7, Massimo, Irvine, CA, USA) at the same time. The device was placed on the lips of awake patients or the tongue of anesthetized dogs. Intraoperatively, during the time the patient was anesthetized (T_INDUC_ − T_45_), ETCO_2_ and ET_ISO_ (ePM12VETc/AA, Mindray, Hamburg, Germany) were also monitored. For all animals, anesthesia time, duration of surgery, extubating time, and anesthetic recovery time were recorded.

### 2.5. Analgesic Rescue

During the anesthetic–surgical procedure, when an increase in the sympathetic tone was recorded (e.g., increases of more than 20% in HR, RR, and MAP in comparison to baseline values) [24,25], fentanyl (Fenodid, Pisa, Mexico City, Mexico) at 5 μg/kg as a single bolus IV was administered as intraoperative analgesia.

### 2.6. Ethical Considerations

The procedures and animal handling during blood sampling were performed in accordance with the Official Mexican Standard NOM-062-ZOO-1999 [26]: “Specifications and techniques for the production, care, and use of laboratory animals”. The study was approved by the Internal Committee for the Care and Use of Experimental Animals (CICUAE) of the Faculty of Higher Studies Cuautitlán. The ARRIVE guidelines were implemented to improve the design, analysis, and publication of research with animals.

### 2.7. Statistic Analysis

The GraphPad Prism statistical package (ver. 10.1.1; Boston, MA, USA) was used to obtain descriptive statistics for all variables. The normality of the data was analyzed by the Shapiro–Wilk test. The results recorded as anesthesia time, duration of surgery, extubating time, and anesthetic recovery time were expressed in minutes as mean ± standard deviation (SD) and presented in a table together with the results of ET_ISO_, SpHb, Hb_LAB_, and in their respective units (mean ± SD) according to the precepts of descriptive statistics.

Hb_LAB_ and SpHb data were analyzed using a Spearman correlation and a Bland–Altman analysis to measure the degree of the statistical linear relationship between the hemoglobin measurement methods, using data obtained in the evaluation corresponding to five minutes before the end surgery, at which time both paired measurements were made. The correlation of SpHb with SpO_2_, SpOC, and PI was calculated using the Spearman method. The cardiorespiratory parameters HR, RR, ETCO_2_, SAP, DAP, MAP, and T° were expressed as mean ± SD. The SpO_2_, PI, and PVi values were expressed as median ± standard error of the mean (SEM). All these parameters were analyzed using ANOVA in a linear mixed model and a Tukey post hoc test. Statistical significance was set at *p* < 0.05.

## 3. Results

A total of 85 dogs were anesthetized; these included 28 mixed-breed dogs, 10 Poodles, 8 Miniature Schnauzers, 6 Labrador Retrievers, 6 Pitbulls, 6 Chihuahuas, 4 Pugs, 3 Beagles, 2 Boxers, 2 Dachshunds, 2 Siberian Huskies, 1 Weimaraner, 1 Pinscher, 1 Maltese, 1 Shiba, 1 French Bulldog, 1 Bloodhound, 1 Bobtail, and 1 Cocker Spaniel. In three of them, SpHb could not be recorded with the co-oximeter, showing a measurement efficiency of 96.47%. No patient required rescue analgesia during the surgical procedure. Table 1 shows the demographic data of the animals under study as well as the average values of anesthesia times, duration of surgery, extubating time, anesthetic recovery time, and values obtained for ET_ISO_, SpHb, Hb_LAB_, and SpOC.

SpHb values were compared with Hb_LAB_. Bland–Altman analysis showed 95% limits of agreement of −4.22 to 4.99 g/dL (Figure 2), with a BIAS (mean difference) of 0.384 ± 2.35 g/dL (Figure 3) (r = 0.401) in patients with adequate perfusion (PI > 0.36) during the anesthetic–surgical procedure. Table 2 shows that the monitored cardiorespiratory parameters were found to be at physiological values typical of an anesthetized patient, showing only a statistically significant difference to their baseline value (*p* < 0.05). SpHb recordings had strong correlations with SpO_2_ (r = 0.995) and SpOC (r = 0.992) and moderate correlation with PI (r = 0.418).

## 4. Discussion

Pulse co-oximetry is frequently used in human anesthesiology as an advanced monitoring tool due to its application in various clinical conditions [27,28]. In contrast, its use in veterinary medicine has been limited, requiring further studies. In the present study, and according to the Bland–Altman analysis, SpHb values recorded a 95% limit of agreement of −4.22 to 4.99 g/dL when compared to Hb_LAB_ values, and the SpHb measurements were lower than the corresponding laboratory measurements. These values show concordance with the results described by Tayari et al. [13], who reported a bias between SpHb and Hb_LAB_ of between 0.3–1.0 g/dL post adjustment in vivo, using an arterial blood sample. Particularly in our study, the determination of Hb_LAB_ was from a venous sample, which can be highlighted as a strength since it represents a less invasive method that requires fewer technical skills compared to arterial sampling. Contrarily, Read et al. [29] reported that co-oximeter-derived Hb did not have good agreement with laboratory-measured Hb in dogs under anesthesia. A possible explanation from a strictly statistical perspective for this effect is related to the use of a larger sample size, which could confer greater power to the analysis presented in this study [30]. This would also help to understand why studies carried out in humans have reported the precision of this technology to determine and monitor SpHb [28,31,32,33].

The present findings suggest the clinical utilization of pulse co-oximetry to measure Hb in dogs. The moderate positive correlation between SpHb and Hb_LAB_ (r = 0.401) suggests clinical agreement between laboratory measurements and records obtained from the monitor, which suggests the clinical application of co-oximetry in dogs. Colquhoun et al. [14] reported a co-oximetry sensitivity of 67–75% and a specificity of 95–84% to determine increases or decreases in Hb of more than or equal to 1 mg/dL, respectively, suggesting that monitoring SpHb through co-oximetry might serve as a guide to estimate Hb blood concentration.

The value of continuous SpHb monitoring should not be based on the premise that this parameter can replace Hb_LAB_ since the greatest benefits of continuous SpHb recording come from providing continuous values of Hb to indicate hemodynamic stability in critically ill or anesthetized patients [34]. In this sense, Ayres [35] mentions that this technological tool can also measure the presence of functional hemoglobins such as carboxyhemoglobin, oxyhemoglobin, deoxyhemoglobin, and methemoglobin, improving the perioperative monitoring of the anesthetized patient in whom pulse oximetry alone is commonly used. A pilot study made by Swann et al. [36] compared co-oximetry and an invasive method in 10 conscious dogs. The authors found an upper and lower agreement of 4.21 and −1.25 g/dL and a correlation of r = 0.065 (*p* < 0.0001), findings that were similar to the present research. Thus, co-oximetry might be a reliable technique for non-invasively measuring Hb in real time in dogs.

SpHb in the present study had a high correlation with the cardiorespiratory parameters SpO_2_ and SpOC, since O_2_ is transported mainly combined with Hb in >98%. SpO_2_ measures the amount of oxygen carried by the blood compared to its total capacity; therefore, perioperative monitoring of these physiological variables is important since it helps to understand the transport of this gas and provides immediate information on the patient’s health status, especially in cases where there is hypoxia or anemia, even in those situations where transfusion is required [8,9].

Studies in human medicine have concluded that pulse co-oximetry equipment uses multiple-wavelength spectrophotometry (500–1400 nm) to measure the concentration of SpO2, SpHb, SpMet, and SpCO. These differences in wavelengths allow the functional saturation of Hb with oxygen to be measured, which is considered an advantage over standard pulse oximetry since the latter cannot detect or differentiate dyshemoglobins [37,38]. This would add greater value to the agreement of the data in SpHb and Hb_LAB_. However, in the present study, SpHb values were slightly lower than Hb_LAB_. These differences could be related to what was observed in horses, where different studies reported that pulse oximetry can underestimate the calculated SaO_2_ by 3.3–3.6%. This variation is attributed to temperature and pH alterations [39,40]. These factors are inherent to the anesthesia period and could be considered a limitation, as also observed by Zoff et al. [40], who suggest that measurement of SpHb in horses cannot be recommended as a substitute for direct Hb assessment and is required to be correlated with SaO_2_ and CaO_2_ as determined by gasometry.

In anesthetized dogs, Tayari et al. [13] determined the importance of performing an in vivo adjustment 15 min from the first reading since before this maneuver, the co-oximeter might overestimate Hb and CaO_2_ values. In contrast, after the in vivo adjustment, the accuracy and precision of SpHb improved, obtaining a difference between both measurement methods of 0.3 g/dL, a finding that was similar to what was observed in the present research. This in vivo adjustment to improve the precision of SpHb has also been reported in human studies with different sensors [41] or during surgeries with a high risk of hemorrhage, where the measurement of SpHb could optimize blood transfusion [6].

Another factor that affects the efficiency of the technique is related to the affinity of Hb with O_2_, as has been reported in horses. In this species, a P50 value of 23.8 ± 0.8 mmHg of the affinity of the O_2_ has been observed [42], a value that is lower compared to what was observed in humans (26.6 mmHg), while dogs have a value of 30.0 ± 1.3 mmHg [43]. Thus, differences in Hb affinity and conformation in terms of species and breed can cause variations in SpHb recordings, meaning a possible limitation of the technique in a clinical setting [40]. In this regard, in the present study, SpHb could not be recorded in three dogs due to the pigmentation of the mucous membranes and tongue. These readings were similar to those reported by Read et al. [29] who were able to obtain 96.9% of anesthetized patients’ values. This also coincides with what has been observed in humans, where the presence of tar, soot, or nail polish hinders reading and in vivo adjustment. Furthermore, individuals’ movement can directly alter the absorbance level of infrared light, which can underestimate or generate a null SpHb reading [37,38].

The present results are similar to some clinical studies conducted in humans. For example, Macknet et al. [44] reported an average difference between SpHb and Hb_LAB_ of −0.15 g/dL. Likewise, Isosu et al. [45] also obtained a bias of −0.7 ± 1.0 g/dL between SpHb and Hb_LAB_ after in vivo adjustment, with limits of agreement of −2.8 to 1.4 g/dL (r = 0.87) in patients with an adequate perfusion index. Another study by Moore et al. [32] showed a variation from 3.89 to −3.84 g/dL; however, their study did not report a significant correlation between SpHb and Hb_LAB_ values. In general, non-invasive assessment of SpHb using the pulse co-oximeter used in the present study has been approved by the United States Food and Drug Administration, reporting a difference of ±1.0 g/dL in Hb when compared to Hb_LAB_ values in adult humans (range between 8–17 g/dL) [28,31,32,33]; these parameters were similar to those observed in the anesthetized dogs of this clinical study under the elective surgery model.

In human medicine, pulse co-oximetry based on multiple wavelengths of light has shown greater clinical effectiveness in polytraumatized patients, during phlebotomies, and for the timely care of perioperative anemias [46,47], which could be studied in the future in veterinary patients. It has also been identified that patients with compromised peripheral perfusion, in a state of shock, and medicated with vasopressors show measurement biases [46,48], factors that were not observed in the present research. The influence of these factors might be due to alterations in peripheral blood flow in situations where global perfusion is compromised or when peripheral vasoconstriction persists. This may cause SpHb to be underestimated. These factors were not observed in this study but must be considered in future studies.

Another possible study perspective is the evaluation of co-oximetry for the early detection of Hb changes in neonates as has been observed in humans, where the measurement of SpHb could help in assisting in neonatal therapy [49]. Similarly, due to the correlation between SpHb and blood Hb concentrations, continuous SpHb monitoring could help to decide whether or not to proceed with blood transfusion in intensive care units [50]. As proposed by Cros et al. [51], integrating SpHb monitoring with the PVi index into a vascular filling algorithm reduced patient mortality by encouraging the performance of early blood transfusions. Therefore, in a surgical setting where blood loss may not be evident or difficult to treat, continuous monitoring of SpHb rather than intermittent measurement may provide earlier decision elements.

## 5. Conclusions

SpHb has a moderate positive correlation with direct blood Hb concentration. According to the results, continuous measurement of SpHb using non-invasive co-oximetry is a reliable and advanced alternative for monitoring Hb concentration in dogs under anesthesia.

## Figures and Tables

**Figure 1 animals-14-00822-f001:**
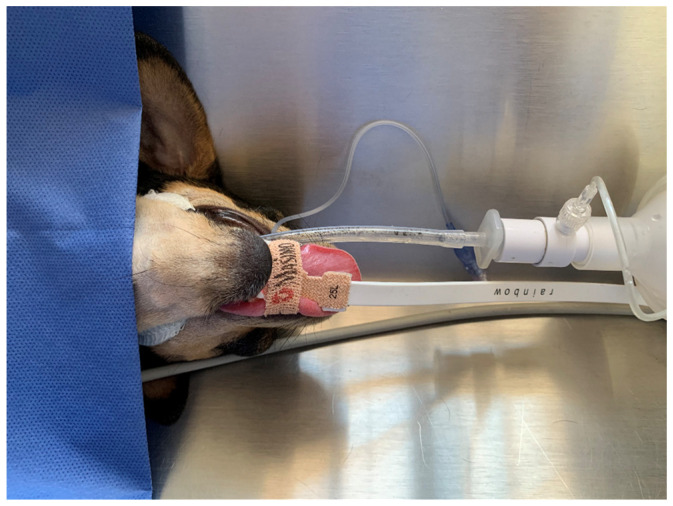
Placement of the adhesive sensor of the pulse co-oximeter in the ventrolateral portion of the tongue of anesthetized patients.

**Figure 2 animals-14-00822-f002:**
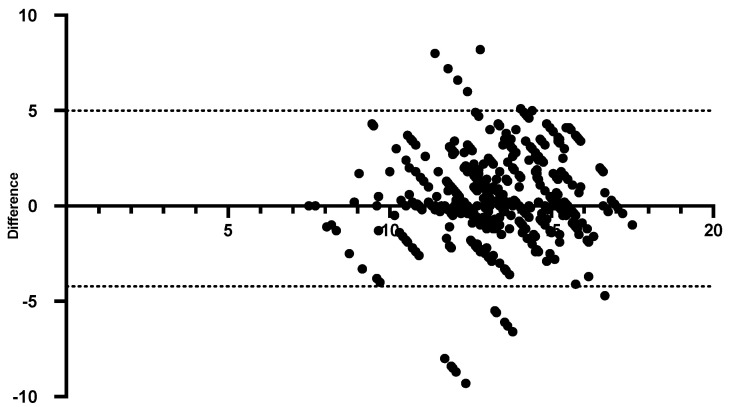
Bland–Altman plot for comparison of SpHb measurements by co-oximetry with Hb_LAB_ (g/dL) after in vivo adjustment.

**Figure 3 animals-14-00822-f003:**
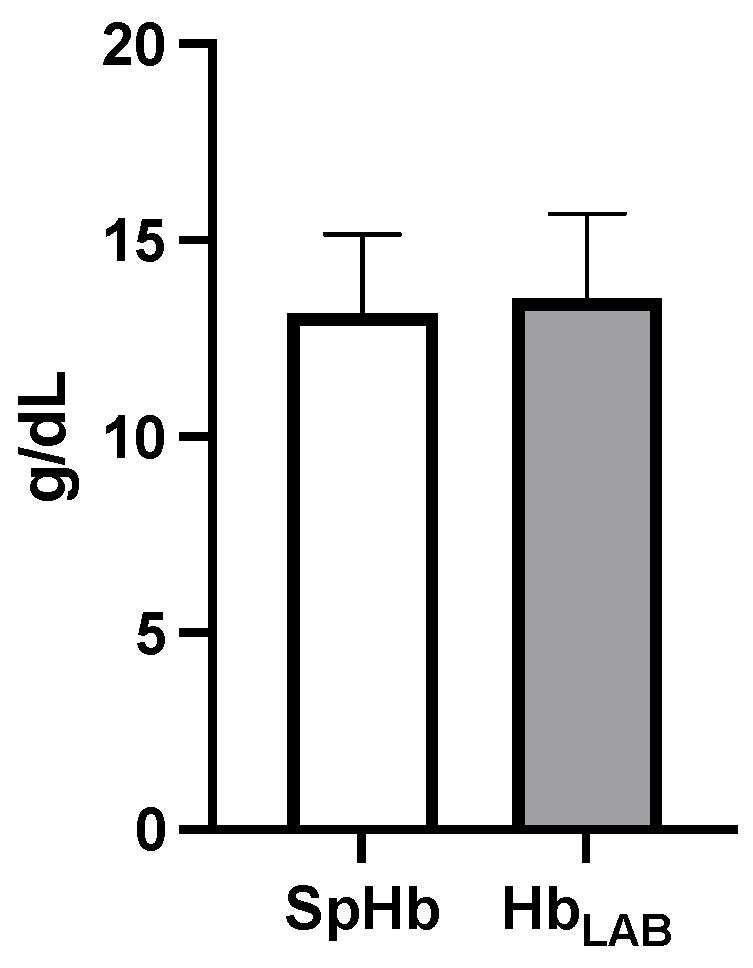
Mean values for SpHb and Hb_LAB._ A BIAS (mean difference) of 0.384 ± 2.35 g/dL was observed between both measurement methods.

**Table 1 animals-14-00822-t001:** Mean ± SD of demographic data, anesthesia time, duration of surgery, extubating time, anesthetic recovery time, ET_ISO_, SpHb, Hb_LAB_, and SpOC.

Parameter	Mean ± SD
Age (years)	4.4 ± 2.9
Weight (kg)	13.4 ± 10.1
Anesthesia time (min)	38.1 ± 11.4
Surgery duration (min)	25.5 ± 10.3
Extubating time (min)	9.3 ± 2.6
Anesthetic recovery time (min)	12.5 ± 4.1
ET_ISO_ (%)	1.57 ± 0.37
SpHb (g/dL)	13.1 ± 2.0
Hb_LAB_ (g/dL)	13.5 ± 2.1
SpOC (mL/mL)	17 ± 3

ET_ISO_: percentage of expired isoflurane during the anesthetic–surgical procedure. SpHb: continuous determination of hemoglobin derived from pulse co-oximetry. Hb_LAB_: blood Hb concentration. SpOC: arterial oxygen content derived from pulse co-oximetry.

**Table 2 animals-14-00822-t002:** Cardiorespiratory parameters during the anesthetic–surgical procedure. ^a,b,c^ Different letters indicate statistically significant differences (*p* < 0.05).

Parameter	T_BASAL_	T_DEXME_	T_INDUC_	T_5_	T_10_	T_15_	T_20_	T_25_	T_30_	T_35_	T_40_	T_45_
Heart rate (HR, beats per minute)	129 ± 24 ^a^	82 ± 30 ^b^	85 ± 27 ^b^	91 ± 32 ^b^	94 ± 25 ^b^	97 ± 21 ^b^	94 ± 21 ^b^	93 ± 17 ^b^	93 ± 17 ^b^	95 ± 18 ^b^	96 ± 17 ^b^	96 ± 18 ^b^
Respiratory rate (RR, breaths per minute)	37 ± 10 ^a^	27 ± 9 ^a,b^	17 ± 6 ^b^	20 ± 8 ^b^	17 ± 4 ^b^	17 ± 3 ^b^	16 ± 4 ^b^	16 ± 4 ^b^	16 ± 4 ^b^	17 ± 4 ^b^	17 ± 4 ^b^	18 ± 4 ^b^
ETCO_2_ (mmHg)			35 ± 3 ^a^	36 ± 4 ^a^	38 ± 4 ^b^	38 ± 5 ^b^	39 ± 5 ^b^	38 ± 4 ^b^	39 ± 4 ^b^	39 ± 4 ^b^	39 ± 4 ^b^	37 ± 4 ^a^
Systolic arterial pressure (SAP, mmHg)	148 ± 40 ^a^	146 ± 39 ^a^	124 ± 32 ^b^	121 ± 25 ^b^	113 ± 25 ^b^	121 ± 29 ^b^	120 ± 27 ^b^	117 ± 32 ^b^	112 ± 29 ^b^	114 ± 33 ^b^	110 ± 26 ^b,c^	109 ± 32 ^b,c^
Diastolic blood arterial (DAP, mmHg)	103 ± 33 ^a^	102 ± 35 ^a^	84 ± 30 ^b^	78 ± 22 ^b^	72 ± 23 ^b^	79 ± 25 ^b^	78 ± 23 ^b^	76 ± 28 ^b^	71 ± 25 ^b^	71 ± 27 ^b^	68 ± 23 ^b^	69 ± 26 ^b^
Mean arterial pressure (MAP, mmHg)	117 ± 33 ^a^	116 ± 35 ^a^	97 ± 30 ^b^	92 ± 22 ^b^	86 ± 22 ^b^	92 ± 25 ^b^	90 ± 22 ^b^	90 ± 29 ^b^	84 ± 26 ^b^	85 ± 28 ^b^	81 ± 23 ^b^	82 ± 28 ^b^
Temperature (°C)	38.7 ± 0.5 ^a^	38.7 ± 0.5 ^a^	38.4 ± 0.8 ^a^	38.3 ± 0.7 ^b^	38.1 ± 0.8 ^b^	37.9 ± 0.8 ^b^	37.8 ± 0.9 ^b^	37.6 ± 0.9 ^b,c^	37.5 ± 1.0 ^b,c^	37.5 ± 1.1 ^b,c^	37.4 ± 1.3 ^c^	37.3 ± 1.3 ^c^
SpO_2_ (%)	95 ± 0.4 ^a^	92 ± 0.6 ^c^	97 ± 0.5 ^b^	97 ± 0.3 ^b^	98 ± 0.3 ^b^	97 ± 0.2 ^b^	97 ± 0.2 ^b^	97 ± 0.2 ^b^	97 ± 0.3 ^b^	97 ± 0.3 ^b^	97 ± 0.3 ^b^	98 ± 0.3 ^b^
Perfusion index (Pi, %)	0.49 ± 0.35 ^a^	0.36 ± 0.32 ^a^	0.37 ± 0.21 ^a^	0.44 ± 0.07 ^a^	0.46 ± 0.36 ^a^	0.47 ± 0.21 ^a^	0.46 ± 0.12 ^a^	0.43 ± 0.14 ^a^	0.44 ± 0.23 ^a^	0.42 ± 0.06 ^a^	0.41 ± 0.06 ^a^	0.48 ± 0.08 ^a^
Plethysmographic variability index (PVi, %)	29 ± 1.6 ^a^	26 ± 1.9 ^a^	20 ± 2.3 ^b^	17 ± 1.3 ^b^	16 ± 1.1 ^b^	18 ± 1.2 ^b^	18 ± 1.2 ^b^	17 ± 1.3 ^b^	15 ± 1.3 ^b^	17 ± 1.5 ^b^	19 ± 1.5 ^b^	17 ± 1.4 ^b^

## Data Availability

Data are contained within the article.

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
