# Peer review of "Correlation of Blood Hemoglobin Values with Non-Invasive Co-Oximetry Measurement of SpHb in Dogs Undergoing Elective Ovariohysterectomy"

_animals, 2024, doi:10.3390/ani14060822_

Round 1
Reviewer 1 Report
Comments and Suggestions for Authors
Generally a well presented and written paper. Effective monitoring during veterinary anesthetic procedures is an important area that has not received sufficient attention in the past.
The main issue to rectify is how the in vivo adjustment is undertaken. If the process followed is based on the referenced papers [12,13,43] then a blood sample would have needed to be taken and analyzed using a direct blood estimation of blood Hb. The actual method/ calculation should be given. Discussion on the affect if this should be included.
Also more explanation of how the single HbLAB is statistically compared to the many SpHb data should be given.
58: "I"
66: "estimate the correct ventilation process" needs to be more directly expressed.
73: ..for deciding on when..
234: It is expected that there would be a close relationship between Hb and O2% and PI in most physiological situations, however it is not clear why you are corelating them to the SpO2. You should explain this.
250: ..has been..
260: What about the applicability?
261-264: re-write - vague.
266-268: re-write - unclear
281-284: re-write - convoluted way to indicate that levels of dyshaemoglobins may have affected the SpHb.
Comments on the Quality of English Language
Generally good. A few "typos" need to be corrected and a few statements to be re-written.
Author Response
Generally a well presented and written paper. Effective monitoring during veterinary anesthetic procedures is an important area that has not received sufficient attention in the past.
Response:
The authors thank the reviewer for the time dedicated to reviewing our manuscript, since based on his observations we have been able to significantly improve our article proposal. We appreciate every comment made. Concerning your first observation, the authors express our agreement since in the last decade anesthetic monitoring has been an area of greatest development in veterinary medicine. Regarding the corrections derived from your review, we inform you that you can find them in the revised and updated manuscript in red for quick location; However, in each response, we will indicate the line number where you can see the correction made.
The main issue to rectify is how the in vivo adjustment is undertaken. If the process followed is based on the referenced papers [12,13,43] then a blood sample would have needed to be taken and analyzed using a direct blood estimation of blood Hb. The actual method/ calculation should be given. Discussion on the effect of this should be included.
Response:
Thanks for your observations. About your comment and based on the foundations of MASIMO SET technology, the emitting photodiode uses seven wavelengths that vary between 500-1400 nm to collect data on blood components from the absorption of infrared light. To optimize the SpHb calculation, the monitor software performs a moving mean with the data obtained by infrared spectrophotometry. In this sense, the manufacturer recommends that the moving mean of three previous readings be used as it is considered to present a reasonable balance between speed of response and accuracy of the measurements (in our study this was carried out 15 minutes after the first reading). Thus, the in vivo calibration of the continuous and non-invasive SpHb measurement device consisted of subtracting the value of the same SpHb measured at a given time from the total hemoglobin value obtained from a blood sample taken from the patient at that time same evaluation moment, which as already indicated was after 15 minutes. With this, a number is obtained called the calibration factor (which can be negative or positive), thus the following SpHb values are corrected with that calibration factor obtained automatically. Therefore, with this information, a brief explanation of the in vivo adjustment method in L162-166 has been added to the manuscript.
Also more explanation of how the single HbLAB is statistically compared to the many SpHb data should be given.
Response:
The authors appreciate your observation. For this approach that you suggest, we comment on the following: Bland-Altman analysis is a graphic tool that allows you to compare two measurement techniques, while at the same time allowing you to evaluate the concordance between two sets of data. Thus, in our study, to measure the degree of linear statistical relationship between the hemoglobin measurement methods (HbLAB and SpHb), only the data obtained in the evaluation corresponding to five minutes before the end of surgery were used to develop the Bland-Altman test in paired samples, because it was the moment in which both measurements were made simultaneously, even though SpHb was a parameter evaluated every five minutes after the in vivo adjustment. This decision was also based on the normality of the data, in addition to the fact that during the experiment we tried to guarantee the homogeneity, control, and repeatability of the methods. Likewise, it must be considered that the clinical research carried out used the same experimental tools with the participation of the same observer, the same measurement instrument was always used under the same conditions, there was always control of the criteria inclusion and exclusion, the anesthetic procedure was always performed by the same anesthesiologist, there was always repetition of measurements during a short time under the same research aims, moreover that the surgeries were always done in the same place by the same surgeon. Finally, also clarifies that there were no complications, such as severe hemorrhage, so there were no cases of intra-abdominal hemorrhage that could generate a variation in the HbLAB measurement. This clarification has been made in L218-219.
58: "I"
Response:
In L58 the word “I” has been removed. Thanks for your observation.
66: "estimate the correct ventilation process" needs to be more directly expressed.
Response:
Thank you for your comment, the wording has been modified in L70-71.
73: ..for deciding on when..
Response:
Thanks for your observation, the suggested change has been made in the manuscript at L77.
234: It is expected that there would be a close relationship between Hb and O2% and PI in most physiological situations, however it is not clear why you are correlating them to the SpO2. You should explain this.
Response:
Thank you for your comment. We have added in the discussion in L301-307, the following paragraph: “SpHb in the present study had a high correlation with the cardiorespiratory parameters SpO2 and SpOC since O2 is transported mainly combined with Hb in >98%. SpO2 measures the amount of oxygen carried by the blood compared to its total capacity, therefore, perioperative monitoring of these physiological variables is important, since moreover to helps to understand the transport of this gas, it provides immediate information on the patient's health status, especially in cases where there is hypoxia or anemia, even in those situations where a transfusion is required”. This paragraph is supported by references 8 and 9.
250: ..has been..
Response:
Thank you for your observation, the change has been made in the manuscript at L266.
260: What about the applicability?
Response:
Thank you for your comment, in L281 this idea has been rephrased, considering clinical use.
261-264: re-write - vague.
Response:
Thank you for your comment, the rewrite of the paragraph (L282-284) has been done.
266-268: re-write - unclear
Response:
Thank you for your comment, the rewrite of the paragraph (L286-288) has been done.
281-284: re-write - convoluted way to indicate that levels of dyshaemoglobins may have affected the SpHb.
Response:
Thank you for your comment, the rewrite of the paragraph (L308-312) has been done.
Reviewer 2 Report
Comments and Suggestions for Authors
Thanks for having submitted this very interesting research paper.
I have reported below just few comments that I hope you will found helpful.
Best of luck
Introduction:
Line 58 I think there is a typo error, Pulsioximetry, this should be pulse oximetry or pulse oximetry.
Line 63: I would rather emphasise a bit more the importance of [Hb] assessment preoperatively as O2 is carried as dissolved in solution for <2% and combined with the Hb for > 98%. (https://doi.org/10.1093/bjaceaccp/mkh033) This may explain the importance of its monitoring during general anaesthesia/surgery particularly on which the risk of blood is high.
Line 80: this sentence need to be reviewed as there are several POCT however only pulse co-ximetry technology is capable to continuously and non invasively measure the [Hb]. I hope it makes sense.
Line 87-89: Comparative studies evaluating the accuracy of pulse co-oximetry in equine and canine patients have been previously published (Read 2016; Tayari 2021-2022; Zoff 2019). I would remove the human study from the intro you have a good amount of veterinary studies.
I would probably add a paragraph about the in-vivo adjustment otherwise the readers that are not familiar with the technology would not understand what you have done during the in-vivo calibration.
I would at this point say that to the date only 1 study (Tayari) has evaluated the accuracy of the Masimo after in vivo adjustment in dog and only in one type of surgery and further investigation are require to confirm or deny these finding in other clinical scenarios.
Line 95: you should add "after in-vivo adjustment"
Line 95: your hypothesis is not very clear as it is written. I would suggest you to put more emphasis on the fact that pulse co-oximetry is a non-invasive technologies that allow [Hb] measuremen.
Something like: we hypothesised that the [Hb] measured via pulse co-oximetry acceptably correlates with the laboratory measurement with the advantage of a continuously monitoring.
M&M
You definitely need to explain how you did the in-vivo calibration, using which [Hb] concentration? the pre-GA blood ? please explain
Line 100: I would report the power calculation it is very easy to perform, you can use the Tayari study where only 39 dogs have been enrolled, so you have more than enough but if you report the power calculation your study would result more strong, believe me!
Line 106: I would say only animals classified as ASAI or II were included in the study as ASA is not anaesthesia risk by definition but the classification of your patient physiological status, the risk of anaesthesia is related to also other factors not related to the animal per se...I hope this makes sense.
Line 117: using the most adequate size of blood pressure cuff, remove 3 or 4.
Line 127: and the tracheal tube was connected to an anaesthetic..
Line 131: I would remove to maintain the MAP of 60-90, just adjusted to maintain adequate surgical anaesthetic depth.
Line 139: please report 35-45 mmHg, 30 is a bit too low.
Line 141: please add trying to maintain normothermia.
Line 150: I may have missed, what is the randomisation of your study? what did you randomise and how did you randomise? to me does not look like that you have done randomisation but I may be wrong.
Line 174: remove the size of the cuff.
Line 188: administered IV? please add
Table 1 recuperation time, what does it mean? please report in the legend of the table. Also report in the legend of your table the meaning of all the acronym used, thank you!
Discussion
Line 253: after that you have reported the difference of your study compare to the Read 2016, you should then report the agreement with the Tayari study even if you have used venous instead of arterial blood sample. This is at your favour as the venous sample is less invasive and requires less technical skills compare to a venous blood sample.
Author Response
Thanks for having submitted this very interesting research paper.
I have reported below just few comments that I hope you will found helpful.
Best of luck
Response:
The authors thank the reviewer for his valuable time spent reviewing our manuscript, since based on his observations we have been able to significantly improve our article proposal. Likewise, we inform you that the corrections derived from your review can be found in the revised and updated manuscript in red for quick location; however, in each response, we will indicate the line number where you can see the correction made.
Introduction:
Line 58 I think there is a typo error, Pulsioximetry, this should be pulse oximetry or pulse oximetry.
Response:
Thank you for your comment, the term in L58 and the rest of the manuscript have been corrected.
Line 63: I would rather emphasise a bit more the importance of [Hb] assessment preoperatively as O2 is carried as dissolved in solution for <2% and combined with the Hb for > 98%. (https://doi.org/10.1093/bjaceaccp/mkh033) This may explain the importance of its monitoring during general anaesthesia/surgery particularly on which the risk of blood is high.
Response:
The paragraph has been modified according to your suggestion (L65-69).
Line 80: this sentence need to be reviewed as there are several POCT however only pulse co-ximetry technology is capable to continuously and non invasively measure the [Hb]. I hope it makes sense.
Response:
Thank you for your comment and to avoid confusion for the reader, the phrase in L85: Currently, the alternative is pulse co-oximeters, has been removed.
Line 87-89: Comparative studies evaluating the accuracy of pulse co-oximetry in equine and canine patients have been previously published (Read 2016; Tayari 2021-2022; Zoff 2019). I would remove the human study from the intro you have a good amount of veterinary studies.
Response:
Thank you for your comment. The change has been made considering your suggestion (L95).
I would probably add a paragraph about the in-vivo adjustment otherwise the readers that are not familiar with the technology would not understand what you have done during the in-vivo calibration.
I would at this point say that to the date only 1 study (Tayari) has evaluated the accuracy of the Masimo after in vivo adjustment in dog and only in one type of surgery and further investigation are require to confirm or deny these finding in other clinical scenarios.
Response:
Thank you for your observation, and in agreement with reviewer 1, an explanation of in vivo calibration has been added to L162-166. On the other hand, we agree with the reviewer's opinion about the limited evidence that exists in veterinary medicine on the use of co-oximetry. That is why we present our proposal using a surgical model of ovariohysterectomy.
Line 95: you should add "after in-vivo adjustment"
Response:
Thanks for your observation, the phrase has been added in L99.
Line 95: your hypothesis is not very clear as it is written. I would suggest you to put more emphasis on the fact that pulse co-oximetry is a non-invasive technologies that allow [Hb] measuremen.
Something like: we hypothesised that the [Hb] measured via pulse co-oximetry acceptably correlates with the laboratory measurement with the advantage of a continuously monitoring.
Response:
Thanks for your suggestion, the change has been made in L100-102.
M&M
You definitely need to explain how you did the in-vivo calibration, using which [Hb] concentration? the pre-GA blood? please explain
Response:
Thank you for your comment. As already mentioned, a brief explanation of the in vivo adjustment has been added (L162-166).
Line 100: I would report the power calculation it is very easy to perform, you can use the Tayari study where only 39 dogs have been enrolled, so you have more than enough but if you report the power calculation your study would result more strong, believe me!
Response:
Thank you for your kind comment, we have added the information regarding power calculation and sample size in L106-108.
Line 106: I would say only animals classified as ASAI or II were included in the study as ASA is not anaesthesia risk by definition but the classification of your patient physiological status, the risk of anaesthesia is related to also other factors not related to the animal per se...I hope this makes sense.
Response:
Thank you for your comment, we have made the change according to your suggestion in L111-112.
Line 117: using the most adequate size of blood pressure cuff, remove 3 or 4.
Response:
Thank you for your suggestion, the blood pressure cuff gauge (L124) has been removed.
Line 127: and the tracheal tube was connected to an anaesthetic..
Response:
Thank you for your comment, we have made the change according to your suggestion in L133-134.
Line 131: I would remove to maintain the MAP of 60-90, just adjusted to maintain adequate surgical anaesthetic depth.
Response:
Thank you for your comment, we have made the change according to your suggestion on L137.
Line 139: please report 35-45 mmHg, 30 is a bit too low.
Response:
Thank you for your comment, we have made the change according to your suggestion on L145.
Line 141: please add trying to maintain normothermia.
Response:
Thank you for your comment, we have made the change according to your suggestion in L146.
Line 150: I may have missed, what is the randomisation of your study? what did you randomise and how did you randomise? to me does not look like that you have done randomisation but I may be wrong.
Response:
Thank you for your observation and to avoid confusion for the reader, the word randomized has been eliminated (L157).
Line 174: remove the size of the cuff.
Response:
Thank you for your comment, the size of the cuff (L185) has been removed.
Line 188: administered IV? please add
Response:
Thank you for your comment, the word IV has been added in L198.
Table 1 recuperation time, what does it mean? please report in the legend of the table. Also report in the legend of your table the meaning of all the acronym used, thank you!
Response:
Thank you for your comments, the suggested modifications have been made.
Discussion
Line 253: after that you have reported the difference of your study compare to the Read 2016, you should then report the agreement with the Tayari study even if you have used venous instead of arterial blood sample. This is at your favour as the venous sample is less invasive and requires less technical skills compare to a venous blood sample.
Response:
Thank you for your comments, the modifications suggested have been made on L269-274.